# Flow Velocity Variations and Surface Change of the Destabilised Plator Rock Glacier (Central Italian Alps) from Aerial Surveys

Francesca Bearzot [1,*], Roberto Garzonio [1], Roberto Colombo [1], Giovanni Battista Crosta [1], Biagio Di Mauro [2], Matteo Fioletti [3], Umberto Morra Di Cella [4] and Micol Rossini [1]

1 Department of Earth and Environmental Sciences (DISAT), University of Milano-Bicocca, Piazza Della Scienza 1, 20126 Milan, Italy; roberto.garzonio@unimib.it (R.G.); roberto.colombo@unimib.it (R.C.); giovannibattista.crosta@unimib.it (G.B.C.); micol.rossini@unimib.it (M.R.)
2 Institute of Polar Sciences, National Research Council, 20126 Milan, Italy; biagio.dimauro@cnr.it
3 Snow-Meteorological Centre, Environmental Protection Agency of Lombardia, 23032 Bormio, Italy; m.fioletti@arpalombardia.it
4 Climate Change Unit, Environmental Protection Agency of Valle d'Aosta, 11020 Saint-Christophe, Italy; u.morradicella@arpa.vda.it
* Correspondence: f.bearzot@campus.unimib.it

**Abstract:** Flow velocities were measured on the Plator rock glacier in the Central Italian Alps using a correlation image analysis algorithm on orthophotos acquired by drones between the years 2016 and 2020. The spatial patterns of surface creep were then compared to the Bulk Creep Factor (BCF) spatial variability to interpret the rock glacier dynamics as a function of material properties and geometry. The rock glacier showed different creep rates in the rooting zone (0.40–0.90 m/y) and in the frontal zone (>4.0 m/y). Close to the rock glacier front, the BCF assumed the highest values, reaching values typical of rock glaciers experiencing destabilisation. Conversely, in the rooting zone the small rates corresponded to lowest BCFs, about five times smaller than in the frontal zone. The Plator rock glacier revealed a substantial advancement from 1981 to 2020 and distinct geomorphological features typical of rock glaciers exhibiting destabilising processes. Given the fast-moving phase, the advancement of both the front line and the front toe of the rock glacier, and the contrasting spatial distribution in the BCFs, the Plator could be considered a destabilised rock glacier.

**Keywords:** Italian Alps; rock glacier; creep; permafrost; morphology; Bulk Creep Factor BCF

## 1. Introduction

Rock glaciers are landforms that form as a result of creeping mountain permafrost [1,2]. In recent years, the study of rock glacier dynamics and their coupling to the changing climate system is receiving increasing attention [3,4].

Since the 1990s, acceleration of rock glacier displacements has been documented in the European Alps. Thermo-hydro-mechanical coupling associated with the transitory availability of liquid water content are the main reasons rock glaciers move rapidly. A number of studies have investigated the connection between air and ground temperature and the flow dynamics of rock glaciers [5–8], while others have integrated flow information with environmental factors like sediment supply dynamics and landform characteristics [1,9,10].

In recent decades, time series of rock glacier movement in the European Alps indicate that the acceleration in permafrost creep is strongly related to the availability of liquid water. Increased liquid water content strongly affects the movement of rock glaciers, influenced on various temporal and spatial scales by changes in air and ground temperatures [4,11]. Snowmelt, liquid precipitation, and subsurface water flow result in decreased cohesion between soil/ice particles and/or increased pore pressure [12,13], creating local situations of rock glacier instability and, in specific topographic conditions, may cause natural hazards

to Alpine communities [11,14]. Rock glacier acceleration and destabilisation are mostly related to the complex combination of positive air temperature anomalies and topographical conditions [7,8,10]. The environmental changes affect the processes of transport, resulting in topographical and kinematic changes, and the existence of the ice and sediments mixture and its properties play an important role in controlling the rheology of rock glaciers [1].

Some studies of rock glacier dynamics concentrate on surface displacements and morphology changes [15–17], whereas some other on their internal structure [13,18,19] and liquid water content [12], trying to understand the factors that force rock glacier responses across different temporal and spatial scales.

Only a few limited studies [1,18–21] examined the rock glacier dynamics and morphology changes using numerical flow modelling, physical approaches, and mathematical formulations.

Recently, a new approach has been proposed for describing rock glacier dynamics [21]. This method combines a plastic model for rock glacier thickness with an empirical creep model for ice-rich debris. The authors introduced the definition of the Bulk Creep Factor (BCF) to investigate the relationship between rock glacier thickness, surface slope, and creep rates. The BCF represents the mechanical properties of the rock glacier material, and it allows the separation of the geometrical and the rheological contributions to the velocity component. Therefore, the determination of the BCF allows the discrimination of different rock glaciers or the analysis of different sectors of an individual rock glacier with respect to their rheological properties [21].

A previous work on the Plator rock glacier (Central Italian Alps, Fraele Valley, Lombardia, Italy) was carried out by Scotti et al. [22] for the period 1981 until 2012 [22]. In that study, automatic tracking was used to estimate the surface displacements over 31 years of the rock glacier, which was undergoing a fast-moving phase with mean velocity of 3.73 m/y and 2.97 m/y for the tongue and the front, respectively.

Here, we used repeated drone surveys to analyse the Plator kinematics and surface changes at annual resolution. Morphological modifications and kinematics were evaluated using an automatic image correlation algorithm between the years 2016 and 2020. Spatial patterns of displacements were compared to the spatial variability of the BCF. Results were then compared to those obtained in other rock glaciers in the Alps characterised by different dynamical behaviours.

The objectives of this study were: (i) to estimate the spatial and temporal distribution of the rock glacier displacement velocity, (ii) to highlight surface destabilisation features, (iii) to analyse the rock glacier surface changes, and (iv) to test the BCF applicability and interpret its spatial and temporal patterns using only remote-sensing data.

## 2. Study Area

The Plator rock glacier is a talus-derived tongue-shaped rock glacier located in a tributary valley of the Fraele Valley in the Central Italian Alps (46°30′59.68″N, 10°16′42.27″E, Figure 1a). The rock glacier is surrounded by the steep Cime di Plator rock wall, consisting of dolostone of the "Plator-Cristallo formation" belonging to the Austroalpine Ortler nappe [22,23].

In 2020, the rock glacier extended from 2320 to 2590 m a.s.l., with a length and width of 590 m and 120–155 m respectively. The area of the rock glacier was around 76 850 m$^2$, with an average slope of 27°. The rock glacier was analysed considering three zones (Figure 1b) identified by Scotti et al. [22]. Zone #3 is located between the front line and scarp #2. Approximatively between 2440 m a.s.l. (scarp #2) and 2485 m a.s.l. (scarp #1), Zone #2 is identified. Zone #1 is the rooting zone of the rock glacier that develops from the upper limit outlined by scarp #1 to 2590 m a.s.l. The grain-size of the surface debris layer ranges from small angular blocks (5–30 cm) in Zone #2 to large boulders (1–5 m) in Zone #3. A typical morphology of an active rock glacier with tension cracks and transverse furrows and ridges is observed in Zone #2 and Zone #3.

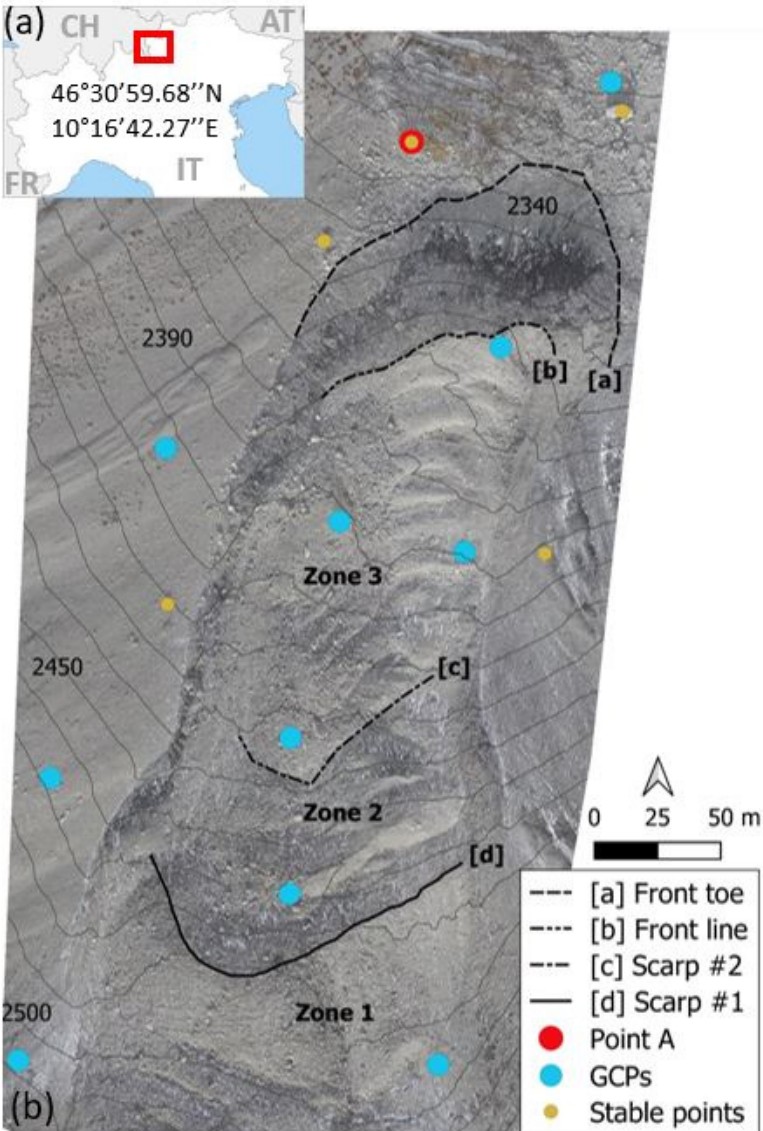

**Figure 1.** (**a**) Location of the study site in the Central Italian Alps. The coordinates of the rock glacier site are given in the WGS 84 coordinates system. (**b**) Orthophoto showing the three zones subdivision, the location of the two scarps [c] and [d], the front line [b] and the front toe [a], the bedrock outcrop and veener in Point A (i.e., position of GNSS/RTK base station, 2337 m a.s.l.), Ground Control Points (GCPs) distribution and five stable points. Image refers to the year 2016.

## 3. Materials and Methods

### 3.1. Data Acquisition

Four drone surveys were performed on 6 October 2016 with a SenseFly Ebee RTK fixed-wing drone, and on 18 July 2018, 29 July 2019, and on 4 August 2020 using a DJI Phantom 4 PRO equipped with a FC6310 RGB digital camera. Photos were shot from a mean distance of 76–82 m above ground level, 3.5–5.0 m/s flight speed (according to light and environmental conditions) and 80% and 75% of front and lateral overlap, respectively. Image resolution (pixels), focal length (mm), and pixel size (μm) were 4896 × 3672 pixels, 4.5 mm for 2016, 5472 × 3078 pixels, 8.8 mm and 2.53 × 2.53 μm for 2018, and 5472 × 3648 pixels, 8.8 mm and 2.41 × 2.41 μm for both 2019 and 2020. Details about the drone campaigns are reported in Table 1.

**Table 1.** Details of the four UAV surveys performed between 2016 until 2020.

| Date | Coverage Area (km$^2$) | Ground Resolution (cm/px) | Flying Altitude (m above Ground Level) | Camera Model | UAV | N° of Images |
|---|---|---|---|---|---|---|
| 6 October 2016 | 0.38 | 4.19 | 157 | DSC-WX 220 | SenseFly Ebee RTK | 52 |
| 18 July 2018 | 0.17 | 2.19 | 81 | FC6310 | DJ Phantom 4 PRO | 272 |
| 29 July 2019 | 0.18 | 2.17 | 82 | FC6310 | DJ Phantom 4 PRO | 207 |
| 4 August 2020 | 0.16 | 1.92 | 71 | FC6310 | DJ Phantom 4 PRO | 300 |

On the rock glacier area, 10 ground control points (GCPs) were evenly distributed (Figure 1b) and their coordinates were measured during drone acquisitions by two GNSS receivers GEOMAX Zenith 35 PRO. The position of the markers was determined using GNSS-Real Time Kinematic (RTK) measurement technique. The base station was positioned at exactly the same location for each drone flight (stable block) in front of the rock glacier (point A in Figure 1b), where its position was determined in post-processing using BORMIO station (BORM) of SPIN 3 GNSS service. For the year 2012, the orthophoto provided by the regional topographic agency of Lombardia Region (Italy), with a pixel spacing of 0.5 m, was used in the analyses.

### 3.2. Photogrammetric Processing

The images collected during each drone survey were processed into orthophotos and Digital Surface Models (DSMs) of the rock glacier and direct surroundings using the Structure-from-Motion (SfM) workflow implemented in the commercial software Agisoft Metashape, v. 1.5.5 [24–26].

The first step in the processing chain was the selection of photographs with sufficient quality. After importing the images, a sparse 3D point cloud was computed by matching coincident features using an image feature recognition algorithm. Successively, an iterative bundle adjustment algorithm was used to construct the 3D geometry and camera position from a sequence of two-dimensional images acquired from multiple viewpoints, and a sparse 3D point cloud was produced. Subsequently, GCP coordinates were imported, and their positions were manually identified within the images to optimise the spatial accuracy of the 3D point cloud. A multi-view stereo image-matching algorithm was used to increase the density of the point cloud and to convert it by interpolation into DSM and orthophoto was derived from the georeferenced image data using the available DSM. The input parameters used in the photogrammetric processing were applied equally for each of the UAV surveys. DSMs and orthophotos were exported with a ground-sampling distance of 0.02 m/pix for each UAV survey.

The Root Mean Square Error (RMSE) and the reprojection error (in pix) were 0.4 cm and 1.2 pix in 2016, 2.0 cm and 0.5 pix in 2018, 3.0 cm and 0.9 pix in 2019, and 2.4 cm and 1.1 pix in 2020.

Five stable ground points outside the rock glacier were used to assess the accuracy of the drone-derived data and the relative kinematic time series. This assessment was carried out by matching features on stable ground, computing x, y, and z shifts for each feature, and followed by then mean and standard deviation. These points were located in stable areas outside the rock glacier, i.e., on lateral talus slopes and on the bedrock outcrop and veneer (close to Point A, see Figure 1b), to also perform a coregistration analysis of each orthophoto and the RMSE were calculated for each time interval investigated.

### 3.3. Determination of Horizontal Surface Velocities

Horizontal surface displacements of the rock glacier were calculated over four time intervals (2012–2016, 2016–2018, 2018–2019, and 2019–2020) using the Correlation Image Analysis System-CIAS [27,28]. CIAS has been already successfully used in other studies focusing on rock glacier dynamics [22,29,30].

CIAS compares two images acquired over the same area at different times to calculate a measure of surface horizontal displacement, using the Normalized Cross-Correlation (NCC) function [22,31,32]. Via block matching, the correlation algorithm searches a reference section in the image acquired at time 1 (t1) in a sub-area of the image acquired at time 2 (t2). The horizontal displacement between the two images was directly given by differences in the central pixel coordinates [27,28,32].

For each rock glacier zone (Zones #1, #2, and #3), morphological features (e.g., blocks) were manually identified in the first orthoimage and automatically recognised from the algorithm in the second orthoimage by their size, shape, and position of each other and then used to evaluate the displacement values. To account for the diverse displacement values and the varying lengths of the periods, CIAS was run with different parameter combinations regarding the block sizes.

Since the deposition of debris at the rock glacier front toe hinders a clear delineation of the front line, this image correlation analysis was not applied. The front and the toe lines were manually digitised on each orthophoto, and their displacements were measured along the most likely rock glacier flow line at each point.

### 3.4. The Bulk Creep Factor and Applicability

The Bulk Creep Factor (BCF) was calculated to investigate the factors controlling the spatial patterns of displacements and to separate the contribution of mechanical properties of the rock glacier material and geometry on the surface velocities [21].

The BCF is defined as the ratio between observed ($C_{obs}$) and modelled creep rates and can be calculated as:

$$BCF = C_{obs} \frac{(n+1)}{\dot{\gamma}c} \left( \frac{\tau_{c\theta} + \rho g H cos\alpha tan\Phi}{\rho g sin\alpha} \right)^n H^{-(n+1)} \quad (1)$$

here $\dot{\gamma}c$ is the critical shear strain rate of the material, $n$ is the flow exponent, $\tau_{c\theta}$ is the cohesion, $\rho$ is the density of the creeping material (given by the contribution of volumetric debris $\omega_d$ and ice content $\omega_i$ and relative densities), $g$ is the gravitational acceleration, $H$ is the thickness of the moving rock glacier, $\alpha$ is the surface slope angle (assuming parallel ground and shear surface), and $\emptyset$ is the friction angle of the shear-zone material.

For Alpine rock glaciers, their thickness can be estimated with a perfectly plastic model [21] using the following formula:

$$H = \frac{\tau}{\rho g sin\alpha} \pm 3.4 \quad (2)$$

Adopting a plastic model for the rock glacier thickness and standard values of the material parameters ($\omega_i = 0.7$, $\rho = 1500$ kg m$^{-3}$, $\emptyset = 25°$, $\tau_{c\theta} = 10$ kPa, $n = 2.1$ and $\dot{\gamma}c = 0.06$ a$^{-1}$) [17], the formulation of the BCF can be approximated to:

$$BCF = 7.6 C_{obs} sin\alpha \left( \frac{0.5}{tan\alpha} + 0.1 \right)^{2.1} \quad (3)$$

For our analysis, BCF values were computed from CIAS dense point measurements and the mean BCF value for each rock glacier zone was calculated and considered representative for that individual zone. Once the BCF was estimated, it was related to the dynamics and geometry of the rock glacier, and the relationships BCF-surface creep rates, BCF-slope angle, and surface creep rates-slope angle were analysed. In addition, for the last analysed period (2019–2020), maps of the spatial distribution of surface flow, slope, and BCF were produced. The surface-creep-rate map was generated by spatial interpolation (Triangulated Irregular Network interpolation method) of the velocity points detected with CIAS to estimate values at other unknown points. Through appropriate mathematical formula (Equation (3)), this product was combined with the slope map of the year 2020 for

creating the BCF map (representation of the spatial distribution of the BCF patterns over the entire portion of the rock glacier) in the 2019–2020 period.

In the absence of information on the internal structure (e.g., thickness of unfrozen debris layer, thickness of frozen core, presence or absence of an unfrozen sediment layer between bedrock, and frozen core and ice content) of the Plator rock glacier, the BCF was calculated using reference values of the parameters as proposed in Cicoira et al. [21]. Furthermore, we assumed a perfect plastic model for rock glacier thickness. This assumption looks realistic for alpine rock glaciers according to the results presented in Cicoira et al. [21], but the validation of this approach should be confirmed by a more detailed dataset.

This method provides information on the zones of the rock glacier that are experiencing a probable destabilisation phase or are set in conditions unfavourable to permafrost preservation. This information may be used to plan detailed field surveys and in-situ measurements for those areas of rock glaciers that are changing and to verify the reliability of the results derived from the BCF analysis, which can be used to understand the long-term evolution of rock glacier dynamics.

## 4. Results

### 4.1. Accuracy of CIAS Estimates

Table 2 reports the horizontal residuals (RMSE xy), the mean values of the maximum correlation and of the average correlation coefficients obtained in the image correlation analysis, and the mean (ME) and standard deviation (STD) of the error computed on stable points for pairs of acquisitions (Figure 1b for the point locations). The horizontal (x and y component) and vertical (z component) accuracy was assessed to evaluate the apparent displacement of stable ground points outside the rock glacier area.

**Table 2.** Root Mean Square Error (RMSE) and the mean coefficients (maximum and average) of stable points derived from CIAS. Mean (ME) and standard deviation (STD) of the horizontal and vertical error for stable ground points.

| Period | RMSE xy (cm) | Mean of Max corr. coeff. | Mean of Average corr. coeff. | ME\|STD x (cm) | ME\|STD y (cm) | ME\|STD z (cm) |
|---|---|---|---|---|---|---|
| 2016–2018 | 0.4 | 0.74 | 0.06 | 0.00\|7.07 | 4.00\|8.94 | 12.00\|4.47 |
| 2018–2019 | 9.8 | 0.87 | 0.05 | 0.00\|0.00 | 0.00\|10.00 | −4.00\|8.94 |
| 2019–2020 | 4.3 | 0.73 | 0.05 | 6.00\|8.94 | 0.00\|7.07 | 4.00\|8.94 |

### 4.2. Velocity Field Distribution

The comparison between drone-derived orthoimages using the CIAS algorithm allowed the horizontal surface displacements at a high spatial resolution to be identified. The number of traced blocks and the mean surface velocity for each zone are shown in Table 3. Results highlighted the complex morphology and heterogeneous creeping pattern of the rock glacier zones. Table 3 shows that displacements did not increase at a constant rate but fluctuated over the measuring period. In Zone #1, the rock glacier moved slowly downslope with a deceleration from 0.7 m/y to 0.4 m/y between 2016 and 2019, respectively. On the other hand, 2019–2020 was the period in which the highest velocities (on average 0.9 m/y) were observed. The surface velocity was significantly higher in Zone #2. Horizontal surface velocity decreased from 2.30 m/y (2012–2016) to 1.40 m/y (2016–2018) and then, after 2018, a gradual increase was observed with a maximum value of around 2.3 m/y reached between 2019 and 2020. Immediately downstream of scarp #2, the largest surface displacements of the entire rock glacier were recorded. Although the trend and the temporal evolution of the movements agreed with those observed in Zone #2, the surface velocities in Zone #3 were about twice as fast. Following a period of reduced displacements in 2016–2018, the rock glacier entered an acceleration phase with velocities increasing from 3.50 m/y in 2016–2018 to almost 4 m/y in 2019–2020.

**Table 3.** Table showing the number of traced blocks used to estimate the surface velocities in Zones #1, #2, and #3. The last two columns refer to the surface velocity of the front line and front toe of the rock glacier. For time intervals 2012–2016 and 2016–2018, the values refer to the annual average.

| Period | N° Traced Blocks | | | Mean Surface Velocity (m/y) | | | | |
|---|---|---|---|---|---|---|---|---|
| | Zone 1 | Zone 2 | Zone 3 | Zone 1 | Zone 2 | Zone 3 | Front Line | Front Toe |
| 2012–2016 | 47 | 75 | 74 | 0.71 | 2.30 | 4.24 | 2.69 | 2.98 |
| 2016–2018 | 141 | 98 | 91 | 0.60 | 1.40 | 3.50 | 3.55 | 3.27 |
| 2018–2019 | 148 | 102 | 107 | 0.41 | 1.70 | 3.84 | 3.61 | 3.18 |
| 2019–2020 | 55 | 38 | 122 | 0.90 | 2.25 | 3.87 | 3.42 | 3.42 |

The downstream movement of the rock glacier caused advancement of the front line and the front toe. The rock glacier front line advanced at a velocity greater than 2.5 m/y following the main rock glacier flow line. In comparison, the presence of a topographic depression associated with a bedrock outcrop and veneer (Point A, Figure 1b) caused the front toe to advance at a rate lower than the average in its western part. The front toe moved at an estimated mean velocity between 3 m/y and 3.5 m/y.

### 4.3. Rock Glacier Morphological Changes

Between 2016 and 2020, the surface morphology of the rock glacier changed markedly (Figure 2) due to its continuous advancement and to its strong spatial variations in flow velocity. Displacement rates were small in Zone #1 while the morphological changes occurred mostly in Zone #3 and at the front where the highest movement rates were found.

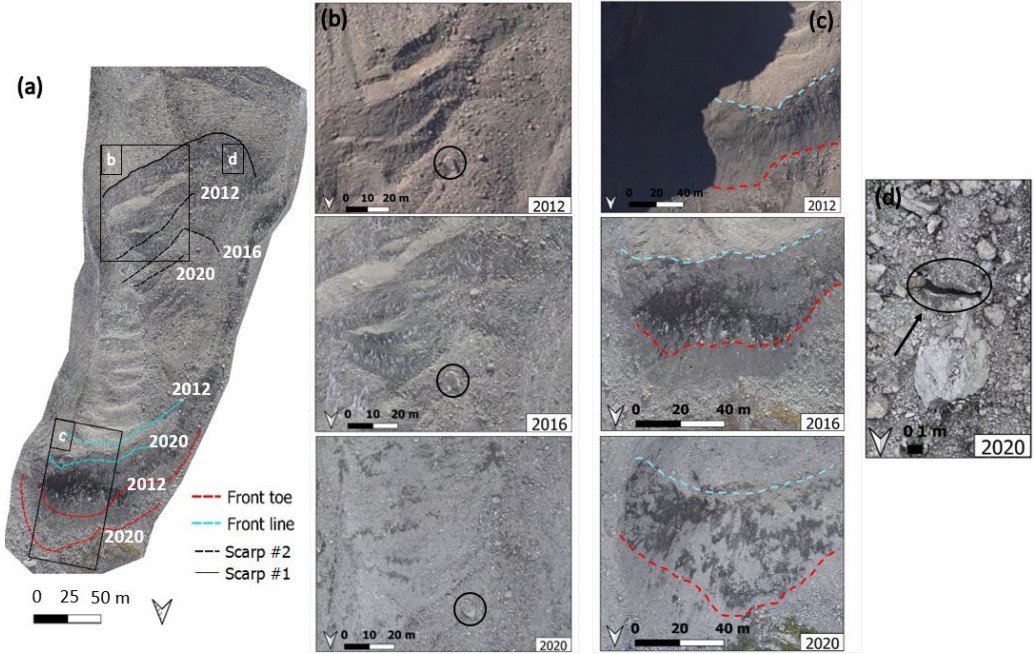

**Figure 2.** (**a**) Morphological evolution of scarp #2 (2012–2016–2020) and advancement of the front line and the front toe (2012–2020) of the rock glacier; (**b**) Surface morphological changes in Zone #2; (**c**) Morphological changes at the rock glacier front; (**d**) Open fissure formation in Zone #2.

The ongoing evolution of the rock glacier surface was indicated by distinct changes in its dynamics and modifications of the surface topography.

The shape of scarp #2 changed significantly (Figure 2a) over the 8-year period analysed in this study (2012–2020) and appeared deep enough to be considered as an important element controlling the rock glacier destabilisation. The scarp evolved considerably from 2012 to 2016 in terms of both length and shape, cutting the entire rock glacier following

the activation of the orographic left-hand side of the rock glacier. On the other hand, in 2020, the development of the scarp was like that of 2012, affecting only one side of the rock glacier. Considering the period between 2012 and 2020, it appeared particularly active and moved downstream constantly changing its morphological appearance. This also caused the separation of the rock glacier into two bodies with different kinematics.

In 2020, Zone #2 (Figure 2b) was characterized by a relatively smooth and unstructured surface compared with previous years, likely due to new sediment input from the neighbouring rock walls.

The rock glacier showed extraordinary changes at the front (Figure 2c) due to the high displacement rates. Due to this, individual debris slide and gravitational movements in the form of isolated collapses randomly occurred. Caused by high velocities and the consequent advance of the rock glacier, subsidence and open fissures developed (i.e., Figure 2d), which are indicative of landform degradation and destabilisation. The fissures were mostly found in Zone #2, downstream of scarp #1, or close to scarp #2.

### 4.4. Spatial and Temporal Variability of the Bulk Creep Factor

Figure 3 shows the surface velocity as a function of the BCF and the surface slope angle for Zones #1, #2, and #3 in the periods 2012–2016, 2016–2018, 2018–2019, and 2019–2020. For the multiannual intervals (2012–2016 and 2016–2018), the values of velocity, slope, and BCF refer to the annual average of the given period.

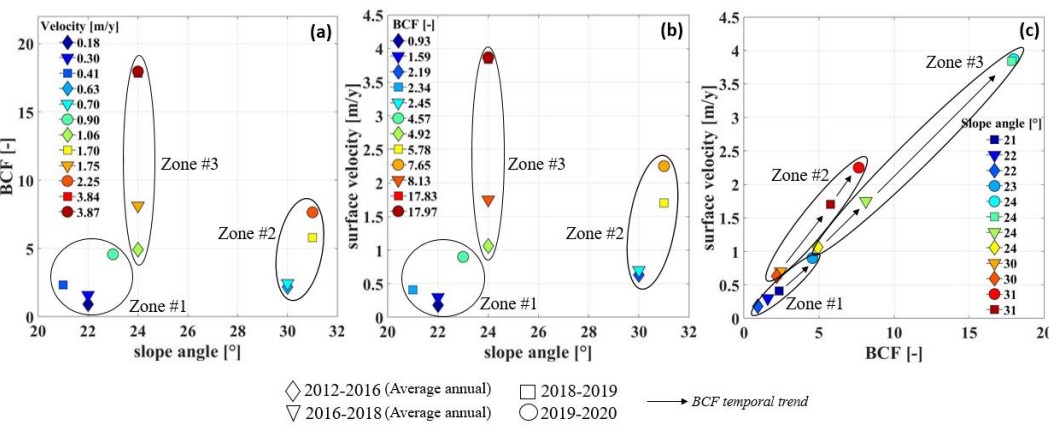

**Figure 3.** Representation of the relation between (**a**) slope angle and BCF; (**b**) slope angle and surface velocity; and (**c**) BCF and surface velocity for Zones #1, #2, and #3.

The maximum calculated BCF values (up to 16) were found in Zone #3 close to the front, where the greatest surface rates were detected. The opposite situation was found in the rooting zone (i.e., Zone #1), where small rates corresponded to the lowest BCF values, about five times smaller than in the frontal zone.

Zone #1 was the most gently inclined, with a slope angle between 21° and 23°, and showed the lowest surface velocities (<1 m/y), resulting in the lowest BCF values, in the range 0.9–4.6. Zone #2 was characterized by surface creep velocities between 1.4 m/y and 2.5 m/y, BCF values between 2.2 and 7.7, and the highest values of surface slope angle (30°–31°). Surface slope angle of 24° and maximum values of both BCFs (17.8–18) and surface velocities (>3.5 m/y) were representative of Zone #3.

The variability in flow velocities observed in the rock glacier zones can be explained by the BCF values (Figure 3c), but a unique relation with the surface slope angle (Figure 3b), which in turn is not directly related to the BCFs (Figure 3a), could not be found.

According to the results, since there were no significant differences in the average slope angle between Zone #1 and Zone #3, the different surface velocities can be expressed by variations in the BCF. On the contrary, surface velocity of Zone #2 was driven by variations in both the BCFs and the slope angles.

Generally, a decrease in flow velocities corresponds to a decrease in the BCF values and vice-versa. Indeed, the velocity patterns depend on the high spatial discontinuity in the BCF between the rooting zone (Zone #1) and the lower part (Zone #3). Hence, proceeding from Zone #1 to Zone #3, the creeping process increased with the BCFs, reaching values typical of rock glaciers experiencing destabilisation behaviour. Furthermore, between 2012 and 2020, the mean BCF value representative of each zone increased progressively, with a dramatic rise especially in the period 2018–2020 and particularly pronounced in Zone #3. This latter zone also showed much more pronounced annual variability in BCFs and in surface velocities than Zones #1 and #2.

In Figure 4, the spatial distribution of the surface slope angles (Figure 4a), the surface velocities (Figure 4b), and the BCF values (Figure 4c) for the most recent time interval are shown (2019–2020). The rooting zone (Zone #1) is the area where the minimum values of all these three parameters were found. Here, the mean surface velocity was lower than one metre per year, the mean BCF representative of the area was 4.6, and the average slope was 23°. The mean BCF value increased (7.7) in Zone #2 as well as the average slope (31°) and the creep rate, which reached 2.3 m/y. A further increase in surface creep rate was seen in Zone #3, with values of almost 4 m/y, which was also associated with an increase in BCF, almost reaching a value of 18. On the contrary, the average surface slope angle (24°) was lower than that calculated for Zone #2.

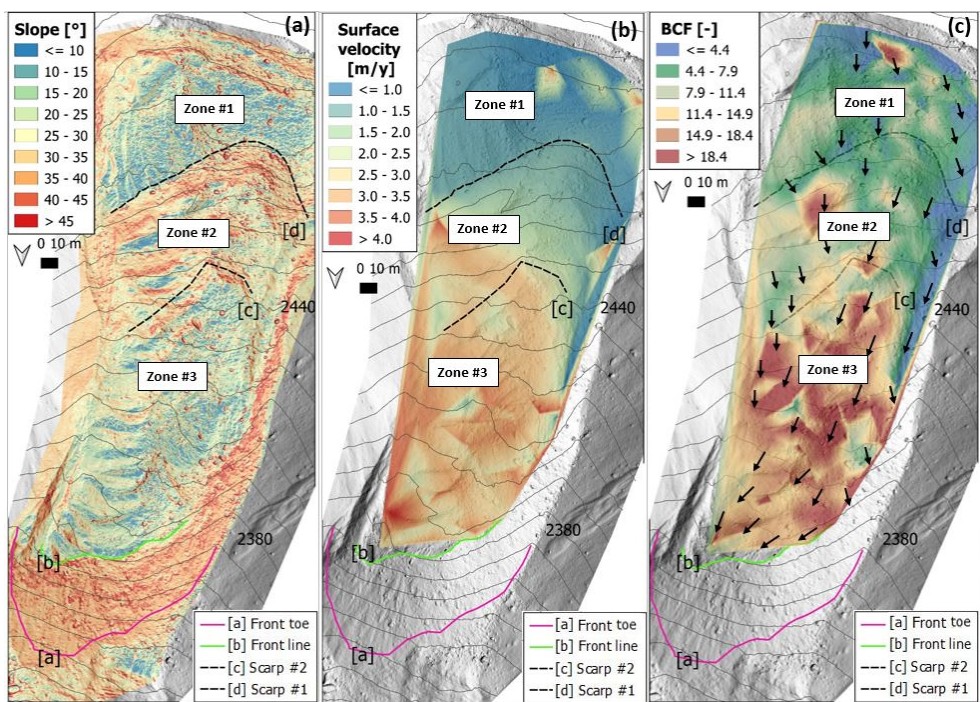

**Figure 4.** (**a**) Surface slope; (**b**) Spatial distribution of surface velocity; (**c**) Distribution of the BCFs and flow direction (arrows) of the surface creep. Period 2019–2020.

The differences in the spatial patterns of surface velocity between the upper and the lower part of the rock glacier can be primarily explained by contrasting and discontinuous rheological properties (BCF) and secondarily by the slope angle differences between Zone #2 and Zones #1 and #3.

### 4.5. Vertical Surface Elevation Changes

Vertical changes (Figure 5a) were detected by the difference of DSM (DoD) between 2016 and 2020, which showed distinct subsidence features of different magnitudes on the rock glacier surface. Figure 5b shows the mean annual vertical variations of scarps #1 and #2 between 2016 and 2020.

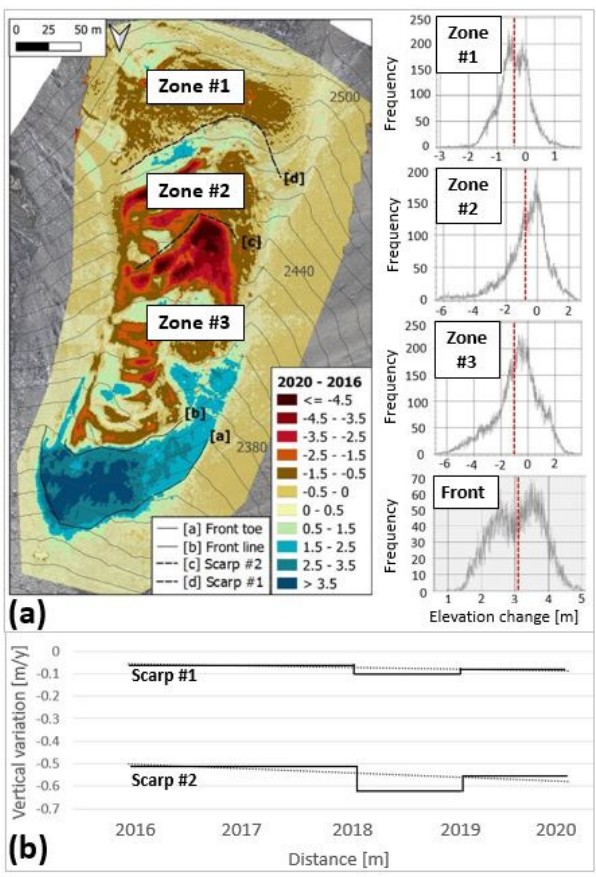

**Figure 5.** (**a**) DSM of differences (DoD) between 2016 and 2020 (left) and the associated frequency distributions of vertical changes (m) of the three zones and of the front part in the period 2016–2020 (right). The red dotted lines represent the mean difference. Please note the different axis scales for the four zones. (**b**) Mean annual vertical variations of scarps #1 and #2 with the respective linear regressions (black dotted lines). Period between 2016 and 2020.

Negative elevation changes could indicate surface subsidence or mass-loss processes, while positive changes may be due to an advancing of the rock glacier front or individual boulders moving across the rock glacier surface. Elevation changes in Zone #1 ranged from maximum values of 1.6 m to −2.4 m. Surface subsidence occurred in the rock glacier over the four-year period of measurement, particularly in the right-hand side of Zone #2 and immediately below scarp #2, where values of more than −4.0 m were reached (i.e., red colour in Figure 5). A major material loss in Zone #2, with negative changes of more than 4.0 m, was recorded, with a maximum peak of 5.6 m. An isolated area with positive changes was located below scarp #1, showing values between 0.4 m and 2.4 m. A slight accumulation of material was mapped in Zone #3 in the advancing ridge and furrow complexes. In these areas, the positive changes reached maximum peaks of 3.6 m, while the most intense material loss covered a range of 2.4 m and 4.4 m with peaks up to 5.6 m. Due to the advancement of the main body, only positive changes occur between the front line and the front toe with values ranging from 0.6 m to over 4.5 m. In addition, the presence of a topographic depression associated with a bedrock outcrop and veneer (Point A, Figure 1b) caused a preferential advance of the rock glacier towards the orographic right-hand side, where the most intense positive variations (above 0.8 m/y) also occurred.

## 5. Discussion

### 5.1. Evidence of Rock Glacier Destabilisation

The rock glacier destabilisation usually takes place on steep slopes as the internal shear stress increases with the slope angle [33]. In destabilised landforms, scarps, crevasses,

and fissures, here called "surface disturbances", can be formed [34]. Surface disturbances are in convex-shape bedrock surfaces where an extensive flow pattern and a thinning of the permafrost body occur [33].

Cases of rock glaciers experiencing destabilisation processes reported significant changes in the evolution of surface disturbances [34]. The frequency and size of these geomorphological characteristics can increase over time, creating growing discontinuities in the deformation and in the creep patterns of rock glacier and promoting additional rock glacier instability [34]. Distinct changes in surface topography have been described for several active rock glaciers in the Alps, indicating the destabilisation of these landforms [18,34,35]. Previous studies on destabilised rock glaciers showed that these landforms feature a wide variety of geomorphological characteristics [10,28]. Even with surface disturbances, rock glaciers can remain stable for decades [10], so the presence of these disturbances is not a sufficient condition for destabilisation to occur. Otherwise, the destabilisation process can be linked to an increase in surface disturbances, as in the case of the Pierre Brune rock glacier, where a crack observable since 1952 evolved to a crevasse in 1970 and further crevasses and a scarp formed after several destabilisation events [34]. In other instances, the surface perturbations on destabilised rock glaciers created a discontinuity in the flow pattern model [22,36], but in two rock glaciers analysed by Schoeneich et al. [37] and by Roer et al. [10], high displacement rate (around 2 m/y) is not considered a necessary feature [10,37]. In fact, even rock glaciers that are not subject to an intense acceleration typical of destabilising phenomena and are therefore characterised by low displacements rates may still present cracks, crevasses, and scarps [35].

The surface morphology of the Plator rock glacier is naturally shaped according to spatially variable flow patterns. The top layer of the rock glacier exhibits ridge and furrow topography attributed to compressive flow during the rock glacier creep. Increasing flow velocity from the rooting zone towards the front and recent morphological changes in surface disturbances indicate that the strain rates increased significantly, suggesting an ongoing permafrost body split and/or thinning process, and promoting a possible destabilisation phase [38]. An example is the development of scarp #2, which was already particularly active between 2007 and 2012, as reported by Scotti et al. [22]. Since then, the scarp has shown a continuous morphological evolution over time reinforcing the belief that it could be a shear plane driving the destabilisation of the rock glacier, leading to an abrupt division of the surface velocity distributions.

Based on the classification defined by Marcer et al. [35], the Plator rock glacier can be classified as a "suspected or potential destabilized rock glacier" since the surface disturbances are clearly recognisable and evolve in time and the velocity field distribution is discontinuous with sectors moving significantly faster than others, as in the case of Zones #1 and #3.

Moreover, the downstream movement of the rock glacier is accompanied by a general subsidence of the rock glacier surface. The subsidence can be the result of permafrost degradation and the surface lowering may therefore depend on body acceleration, extensional flow, ice melting, and reduced ice, consequently leading to changes in the BCF values [1,11,21]. As shown by the DoD map (Figure 5), the Plator is subject to a general surface subsidence and mass loss processes, particularly in Zones #2 and #3 with values occasionally reaching 1 m/y.

Between 2016 and 2020, zones of subsidence covered about 55% of the rock glacier, of which about 18%, 28%, and 54% were in Zones #1, #2, and #3 respectively. The sector showing only material gains is the frontal area of the rock glacier (71%), due to the downstream movement of the tongue.

### 5.2. The Bulk Creep Factor Interpretation

The Plator rock glacier was compared with other Alpine rock glaciers for which the BCFs, surface displacements, and slope angles are available [21]: Laurichard, Dirru, Furggwanghorn, and Pierre Brune rock glaciers (Figure 6a). For the Plator rock glacier, the

values referring to the period 2019–2020 were considered. The rock glaciers considered in these studies are characterised by velocity rates between 1 m/y and 6 m/y on relatively steep slopes, with maximum values of 28 degrees. The mean rock glacier altitude varies between 2450 m a.s.l. and 3600 m. a.s.l. The BCF values range between 5 (Laurichard rock glacier) and 23 (Pierre Brune rock glacier). At present, most rock glaciers in the Alps are characterised by increased degrading permafrost conditions due to global warming, but only rock glaciers currently experiencing destabilisation phenomena or set in conditions unfavourable to permafrost conservation are characterised by high, non-constant, and discontinuous BCF values [21].

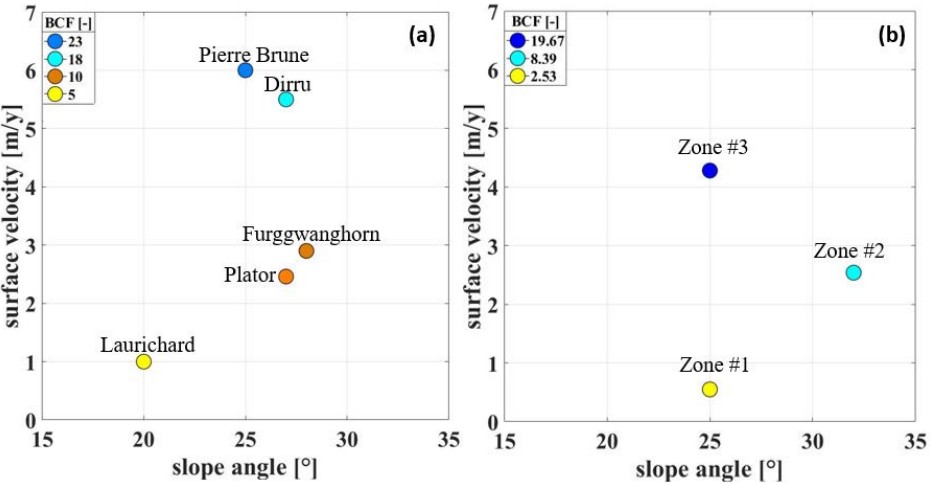

**Figure 6.** (**a**) Representation of the Plator rock glacier together with other rock glaciers (Pierre Brune, Dirru, Furggwanghorn, and Laurichard). The position of the individual rock glaciers in the graph depends on the values of the relationship BCF, slope angle, and surface velocity; (**b**) Representation of Zones #1, #2, and #3 of the Plator rock glacier considering BCF, slope angle, and surface velocity in the period 2019–2020.

Despite its high BCF value, the Dirru is regarded as a rock glacier with very fast flow but not at risk of destabilisation due to almost spatially constant BCF values. The Laurichard rock glacier also shows homogeneous BCF values but is characterised by lower creep rates and slope variations. In both Dirru and Laurichard rock glaciers, there is an absence of abrupt changes in the BCF spatial variability even if the absolute BCF value is different, indicating rheological differences (material properties) and suggesting that the spatial variations in the creep rate cannot be explained by considering the slope angle only.

Both the Pierre Brune and the Plator rock glaciers can be considered as "ongoing destabilisation rock glaciers", although with dissimilar BCF values. They both have distinct BCF patterns between the upper and the lower parts with the highest creep rates observed at the fronts in correspondence of the highest BCF values. Spatial heterogeneity of both velocities and BCF indicates degradations conditions in the rock glacier. With the same mean BCF value of the Plator, the Furggwanghorn rock glacier is also subject to degradation, with acceleration and deepening depression in the rooting zone [18].

For the Pierre Brune, Plator, and Furggwanghorn rock glaciers, the velocity distribution could be explained by the (i) great spatial variation of the BCF, (ii) local topography and morphology (e.g., slope and thickness), and (iii) intrinsic characteristics such as the presence of ice bodies or frozen ground conditions.

On the Plator rock glacier, the heterogeneous velocity patterns between the upper (very low velocity, <1 m/y) and the lower (high velocity, up to >4 m/y) parts combined with high and discontinuous BCFs throughout the entire landform and the continuous development of scarp #2 are clearly signs of ongoing destabilisation processes. The results obtained in this case study are in agreement with those of Cicoira et al. [21] in which

it was stated that rock glaciers currently involved in destabilisation processes or set in unfavourable permafrost conditions show high and discontinuous values of BCF.

Considering the individual zones of the Plator (Figure 6b), Zone #1 evidenced a behaviour comparable with the Laurichard rock glacier, which is described as steady-state creep [21]. Zone #2 showed an increase of the BCF values in the years analysed, suggesting an increase in the rock glacier activity. Zone #3, instead, had BCF values like those estimated for the Dirru rock glacier but differed in the BCF spatial distribution. Zone #3 is composed by a material prone to deformation (high and discontinuous BCF) and is currently experiencing destabilisation processes.

The increase in the BCF is therefore a consequence of a current increase in the deformation state, probably dictated by an increase in ground temperatures, decrease in material cohesion, and water content in the initial step of permafrost degradation [21,38,39]. From 2012 to 2020, the BCF value representative of each Plator zone increased, according to the increase in surface velocities. The Plator rock glacier is undergoing a fast-moving phase typical of destabilised rock glaciers, with a very high surface creep rate in the front zone and a complex spatial pattern of BCF values.

### 5.3. Flow Variations from 1981 to 2020 and Probable Causes of Destabilisation

The actual surface morphology of the rock glacier is the result of the evolution of the rock glacier kinematics and dynamical behaviour throughout the years. Combining the information obtained in this study with the results of Scotti et al. [22], a time interval of 39 years (1981–2020) was covered (Figure 7a).

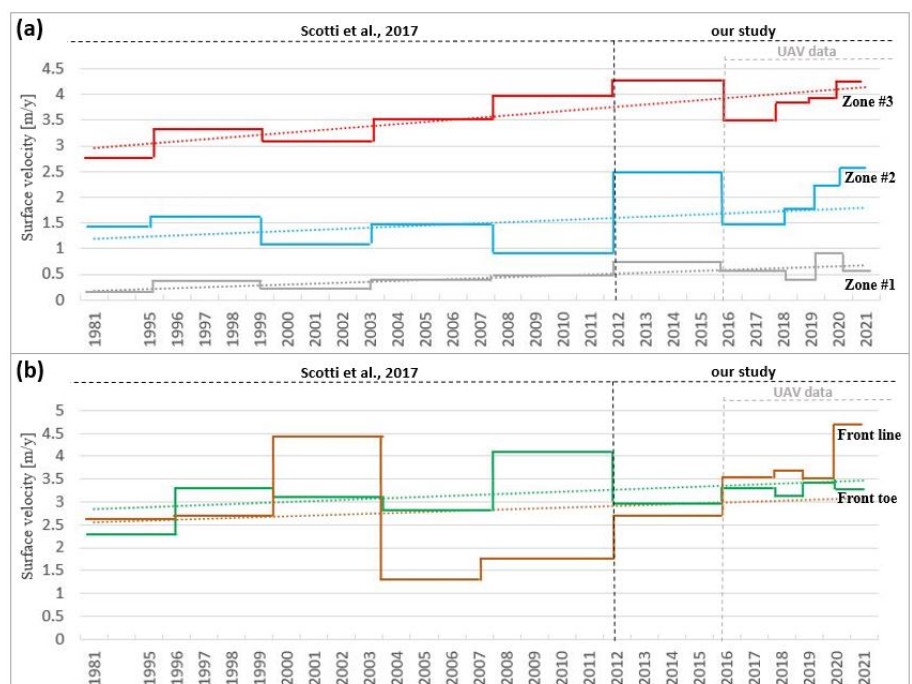

**Figure 7.** Horizontal surface velocities from 1981 to 2020 for: (**a**) Zone #1, Zone #2, and Zone #3; (**b**) Front line and front toe. The coloured dotted lines indicate the linear regression for each time series.

In the rooting zone (Zone #1), the displacement rates varied from 0.2 m/y in 1981 to 1.0 m/y in 2020. In the middle sector (Zone #2), creep rates become more relevant. In the 1981–2012 period, the Plator rock glacier moved downstream at an average horizontal velocity of 1.7 m/y, while in the period 2012–2020 the displacements were higher, reaching velocity higher than 2.2 m/y. Zone #3 reported the highest horizontal surface velocities with values gradually increasing over time. The minimum velocity was recorded in 1981

(with 2.8 m/y) and the maximum peak in 2012–2016 with 4.3 m/y. Between 2016 and 2018, the velocity dropped to 3.5 m/y, but since 2018 there has been a gradual and progressive increase in surface flow velocity.

The rate of advance of the front line and the front toe (Figure 7b) is related to the creep velocity of the tongue. The front toe advanced at 2.3 m/y between 1981 and 1994, increasing up to 4.1 m/y between 2007 and 2012. Between 2012 and 2016, the surface velocity decreased (2.9 m/y), but in the period 2016–2019, a fluctuating trend was observed, ending with an upturn in velocity of up to 3.5 m/y. Concerning the front line, the maximum horizontal surface velocity was recorded in 2003 (4.5 m/y), followed by an abrupt decrease in velocity until 2007 (1.3 m/y). Over the next 12 years (from 2007 to 2019), a steady and progressive increase in creep velocity was observed, reaching a value of 3.6 m/y in 2019. In 2020, the velocity slightly decreased, reaching values comparable to those measured on the front toe.

Some cases of collapse events (as the case of two rock glaciers in South Tyrol [40], and the Bérard rock glacier [41]) and a significant acceleration of rock glaciers have been documented in the European Alps [1,6,36,42] and elsewhere [43,44]. As documented in the Swiss Alps, for many rock glaciers a velocity peak was recorded in 2015, followed by a drop in surface creep rates in 2016–2017, due to ground surface temperature variations that reflect the variations of the temperature of the surface debris layers [8,42]. Then, a continuous increase in surface velocities has been observed since 2018. This temporal course of surface creep velocity can also be observed in the Plator rock glacier, and Zone #3 also showed surface velocity rates comparable to those measured at the Bérard rock glacier (around 3.3 m/y) before its collapse in 2006 [41]. During the general acceleration phase between 2012 and 2015, the Plator showed surface velocity values higher than other rock glaciers [45,46], but some rock glaciers in Valais Alps are instead characterised by markedly higher velocity values, between 3 and 10 m/y [4]. Eriksen et al. [44], studying a rock glacier complex in northern Norway, measured velocities significantly higher than those at the Plator, reporting an increase in the average annual horizontal velocity from 3.6 m/y (2006–2014) to 4.9–9.8 m/y (2009–2016) [44].

Several factors need to be considered to explain the onset and development of rock glacier destabilisation. The development of cracks and the destabilisation of rock glaciers tongues depend primary on the rheological properties of warming ice, while the influence of liquid water in frozen material might be the major factor for permafrost close to 0 °C, and air temperature is an important factor controlling rock glacier speed [10,38]. Kääb et al. [38] demonstrated that rock glaciers characterised by ground temperatures close to 0 °C move usually faster than colder ones because the permafrost creep close to 0 °C is more sensitive to thermal forcing compared with the colder one [38]. Within degrading permafrost, surface velocities increase, the ice content decreases, and the effect of liquid water influences deformation processes [19]. The combination of gravity-driven flow, topography (e.g., surface and bedrock slopes of the rock glaciers), and ice phase creep susceptibility may lead to an increase in deformations, changes in dynamics, and possible development of rock glaciers destabilisation in some cases [19,33,35,47,48].

On the Plator, a clear shear plane is represented by scarp #2, which could drive the destabilisation of the rock glacier, as already pointed out by Scotti et al. [22]. This scarp appeared very active in the period analysed (2012–2020), initially affecting only the western portion of the rock glacier (in 2012) and later (2016–2018) partially developing also in east portion where deep tension cracks were mapped (Figure 2d). Scarp #2 caused the separation of the rock glacier into bodies with different kinematics.

The variations in the rock glacier kinematics can facilitate the development and the creation of surface openings (e.g., fissures) in areas subject to extensional regime, promoting possible increases in the rates of deformation. In addition to high displacement rates, the water input (from precipitation, snowmelt, thawing of the active layer or permafrost, or groundwater flow) within the system can also decrease the cohesive strength between the ice and debris particles [39]. Surface openings can promote the penetration of heat into the

permafrost body and increase the permeability of the rock glacier to both surface water and external temperature, triggering positive feedbacks of rock glacier degradation [35,38]. On the Plator rock glacier, such surface openings caused portions of pure ice to emerge which, without an adequate debris cover layer, are subject to the action of the external temperature, accelerating the degradation process.

High surface displacement rates at some rock glaciers also cause the frontal slope to steepen, increasing the shear stress on the sediment particles on the front [39]. As a consequence, gravitational movements may arise for particles not cemented with ice and beyond a certain threshold. This could be the case for the frontal part of the Plator rock glacier, where random changes in form of small collapses, individual debris slides, or gravitational movements occurred, resulting in local destabilisation phenomena over the years.

The Plator showed an acceleration trend, like many other Alpine rock glaciers, which could also have been favoured by the permafrost warming [49], given the low elevation of its tongue, around 2370 m a.s.l. Heterogeneous horizontal surface velocities between the upper and lower parts of the rock glacier combined with high and contrasting BCF values and the continuous development of scarp #2 are clearly signs of initial destabilisation phenomena.

After this ongoing destabilisation process (high and heterogeneous spatial distribution of the BCF), we expect the BCF of the Plator to drift towards higher values in future years, continuing the growing trend recorded over the years investigated (2012–2020).

However, it is not possible to identify which of these factors play a key role in the destabilisation process of the Plator rock glacier. In the future, more in-depth analyses such as the installation of ground surface temperature sensors and detailed geophysical prospections will therefore be required to better understand which factors play a key role in the ongoing destabilisation process.

## 6. Conclusions

The horizontal surface velocities of the individual zones of the Plator rock glacier were investigated using image correlation analysis and successively discussed in relation to the BCF, which can be used to interpret long-term evolution of the dynamics of the rock glacier. Subsequently, based on the interconnections between BCFs, slope angles, and creep rates, the rock glacier was compared with other Alpine rock glaciers. From the spatial and temporal distribution of the BCF values, and recently formed fissures, the Plator appeared in an increasingly marked state of destabilisation that will tend to continue in the next years.

From 2012 to 2020, the average BCF value of different Plator zones increased due to increasing flow velocities. The high BCF values (close to 18) in the toe zone and the discontinuity of the BCF between the rooting and the frontal zone indicated that the Plator is experiencing a destabilisation phase or set in conditions unfavourable to permafrost preservation.

Like many other rock glaciers in the Alps, the Plator experienced a general increase in surface velocities in the last years. On the Plator rock glacier, surface velocities steadily increased from 1981, reaching values up to 4 m/y in 2019–2020, confirming a trend already observed in the past on the same rock glacier. Increased horizontal surface velocities, development of new scarps, and spatial-temporal BCFs distribution are factors suggesting an ongoing permafrost body split or thinning process.

The Plator is undergoing a fast-moving phase with a very high surface creep rate in the front zone and simultaneously shows high and complex spatial pattern of BCF values, typical of destabilised rock glaciers.

Rock glacier velocities have significantly increased since the 1990s, suggesting that a warming climate may play a key role in this process. The continuous increase in displacement rates will probably lead to further changes and deformation, promoting possible future acceleration and destabilisation events.

The approach shown in this case study is a valuable method for investigating the state of rock glacier activity starting only from remote-sensing data and allows us to highlight rock glacier zones subject to destabilisation processes. This research can be replicated and applied to other landforms such as glaciers and landslides.

**Author Contributions:** Conceptualization, F.B., R.G., U.M.D.C. and M.R.; methodology, F.B., R.G. and M.R.; software, F.B.; validation, F.B. and M.R.; formal analysis, F.B.; investigation, F.B., R.G., G.B.C. and M.R.; resources, U.M.D.C.; data curation, F.B., R.G., G.B.C., B.D.M., U.M.D.C. and M.R.; writing—original draft preparation, F.B.; writing—review and editing, F.B., R.G., R.C., G.B.C., B.D.M., M.F., U.M.D.C. and M.R.; supervision, M.R.; project administration, M.R.; funding acquisition, M.R. All authors have read and agreed to the published version of the manuscript.

**Funding:** This research was funded by the Italian MIUR project Dipartimenti di Eccellenza (2018–2020).

**Data Availability Statement:** Data used in this contribution are available upon request to the corresponding author.

**Acknowledgments:** This research is supported by the GEMMA (Geo Environmental Measuring and Monitoring from multiple plAtforms) laboratory of the University of Milano-Bicocca. The authors thank three anonymous reviewers for their valuable comments.

**Conflicts of Interest:** The authors declare no conflict of interest.

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
