# Peer review of "Flow Velocity Variations and Surface Change of the Destabilised Plator Rock Glacier (Central Italian Alps) from Aerial Surveys"

_remotesensing, doi:10.3390/rs14030635_

Round 1

Reviewer 1 Report

Review of Bearzot et al

„ Flow velocity variations and surface change of the destabilized Plator rock glacier (Central Italian Alps) from aerial surveys“

 The study presented by Bearzot et al describes the surface displacements of the Plator rock glacier derived from UAV imagery. The spatio-temporal displacement pattern is then interpreted with respect to BCF and rock glacier topography. The study essentially extends the previous work of Scotti et al 2016, which used a similar approach.

Although the technical sections of the paper are sound and well written e.g. UAV photogrammetry section and subsequent calculation of the rock glacier spatio-temporal displacement field, the interpretation with respect to BCF is problematic.

The method used assumes a perfect plastic model to determine the thickness of the rock glacier. The input values for H require inputs for t (cohesion) and r (density), however neither of these values are stated in the paper. The fundamental problem lays in the use of a model with static(?) parameters, rendering the calculation of BCF, only with respect to a (slope gradient). Furthermore, BCF values are directly dependent on the velocities ascribed for a given time interval and slope gradient. It is therefore not surprising that BCF values increase with increasing velocity.

Section 4.4 Spatial and temporal variability of the Bulk Creep Factor, lines 339-340:  “….The velocity patterns can be explained primarily by the strong spatial discontinuity in the BCF….” - How does the BCF explain this? Why is the velocity pattern(s) explainable by the BCF and not simply by the spatial distribution of the observed velocity?

Line 41-342: “…(more precisely by the variables and factors on which this depends)….”. Which factors and variables are the authors referring to here? The model does not consider any dynamic change to factors and variables - the authors need to address this further in the text.

The relationship of a BCF value threshold, indicating that the glacier, at some point in time, becomes destabilized, requires further discussion in the paper. What exactly does destabilized mean and how is it defined? If ridge and furrow structures are taken as an indication that a rock glacier is destabilized, it appears that in 1954 (images from Scotti et al) that ridge and furrow features were already formed/forming.

In the absence of a stability analysis (requiring real material parameters and a sound methodology for a rather complex material) what would be the argumentation for a destabilized state?

The authors need to significantly improve the discussion of BCF method application and limitations as to how it directly can explain the displacement patterns observed. Otherwise, I suggest leaving the BCF part out all together and discuss the displacement field pattern(s) along with the geomorphic evidence.

Line 14: “Since the 1990s”

Lines 14-16: The opening statement is quite relevant for this paper, however the message is somewhat misleading. The main reason why rock glaciers move more rapidly has to do with the hydro-mechanical coupling associated with the transitory availability of liquid water. This doesn’t necessarily imply that water is constantly increasing over time because of increasing air and ground temperatures.

Line 42-43: In the absence of knowing what is happening in the subsurface of a rock glacier, the correlation between air temperature and displacement is superficial. A more accurate description would be that the availability of liquid water strongly affects rock glacier movement which is influenced on various temporal and spatial scales relating to changes in air and ground temperature.

Line 56-59: “Only a few limited studies…..” The few studies that consider more than just surface morphology/change are in fact the most significant in trying to understand factors that force rock glacier response across different temporal and spatial scales. The formulation of the text tends to de-emphasize their importance. I would suggest placing the references to the types of study in close association to their outcomes e.g. those reviews are addressed in lines 46 through to 54. A reformulation of the paragraph from lines 42 through to 59 is therefore suggested.

Lines 60-68: BCF is a method for susceptibility assessment in the absence of necessary details e.g. sub-surface data. I suggest the authors address the limitations of the BCF approach in this section.

Line 70: Is the (format) reference to Scotti et al., (2017) correct, given that the reference is designated as 22 according to the MDPI referencing format?

Lines 94-96: Suggested text edit “In 2020, the rock glacier stretches extended from between 2590 to 2320 2320 to 2590 m a.s.l., and it is approximately  with a length and width of 590 m and 120-155 m respectively. long, and width wide with an. The area of the rock glacier was around 76 850 m2, with an The average slope gradient was of 27°, covering an area of around 76 850 m2”.

Line 126: “The base station was positioned at exactly the same location for each UAV flight placed on known position (stable block) in front of the rock glacier (point A in Figure 1b) where its position was determined in post-processing using the SPIN 3 GNSS station of BORMIO (BORM)”.

Line 128: Redundant, therefore delete “The base station was positioned exactly in the same location for each UAV flight”.

Line 129: Suggested text edit “Another product used in the study was the…..”

Section 3.4  The Bulk Creep Factor and applicability:

Line 189: Change (Cobs) to (Cobs).

The following appears contradictory:

Line 339: “These results reinforced the concept that the large variations in surface velocities cannot be explained by the topography and slope angles only”.

Line 565: “….with consequent destabilization phenomena associated, and by the topographical predisposition of the landform. Both factors may have contributed to this fast-moving phase of the Plator, resulting in high and contrasting BCF values in the frontal part (Zone #3), typical of destabilized rock glaciers”

Line 579: “….and the BCF values which allowed to identify zones at risk of destabilization”. How is this statement justified?

Reviewer 2 Report

see  separate sheet

Reviewer 3 Report

The work is very excellent using UAV to character the feature of rock glacial. The velocity rates variability was estimated using the orthophotos, acquired by Unmanned Aerial Vehicle (UAV), between the years 2016 and 2020. Author interpreted the spatial patterns of surface creep through the Bulk Creep Factor (BCF) to investigate the rock glacier dynamics. In addition, author give more explanations of the results and detailed discussion. However, there are still some issues need to be addressed before this paper would be accept

Major comments:

Please do not do that one sentence was set as one paragraph. Author need to reorganize the manuscript’s structure.

Please put high quality of the figure in this manuscript. Most of figures are fuzzy. Particularly, Figs. 1, 4, 5

Minor comments:

  1. Please do not put any references in Abstract section.
  2. The second paragraph is very short. Please consider to combine the first and second paragraph in lines 89-96.
  3. Please improve the quality of the Figure 1
  4. Lines 114-121: This information should be summarized in a table to give a clear data information
  5. Lines 129-130: Please do not use a one sentence as a paragraph
  6. Give me more detail introduction information about the plastic model.

7.Line 215: Clarify the spatial interpolation used in this study.

  1. Can you show which point outside of rock glacier area were used to evaluate the accuracy. For example, marked these points in figure 1.
  2. The results in figure 5 is not clear.
